# Depressive Symptoms, Alcohol Beliefs and Heavy Episodic Drinking in Adolescents

**DOI:** 10.3390/children9010103

**Published:** 2022-01-13

**Authors:** Robert J. Wellman, Catherine M. Sabiston, Matthis Morgenstern

**Affiliations:** 1Department of Population and Quantitative Health Sciences, Division of Preventive and Behavioral Medicine, UMass Chan Medical School, 368 Plantation Street, Worcester, MA 01605, USA; Robert.Wellman@umassmed.edu; 2Faculty of Kinesiology and Physical Education, University of Toronto, 55 Harbord Street, Toronto, ON M5S 2W6, Canada; catherine.sabiston@utoronto.ca; 3Institute for Therapy and Health Research, IFT-Nord, Harmsstrasse 2, 24114 Kiel, Germany

**Keywords:** depression, drinking attitudes, drinking motives, binge drinking, longitudinal study, adolescent

## Abstract

Adolescents who engage in heavy episodic drinking (HED—i.e., 5+ drinks on a single occasion) increase risks for psychopathology, alcohol dependence, and similar negative consequences in adulthood. We explored associations among depressive symptoms, positive alcohol beliefs, and progression of heavy episodic drinking (HED) in 3021 German adolescents (M(SD) age at baseline = 12.4 (1.0)) followed for 30 months in 4 waves, using a conditional parallel process linear growth model, with full information maximum likelihood estimation. By wave 4, 40.3% of participants had engaged in HED more than once; 16.4% had done so ≥5 times. Depressive symptoms were indirectly related to baseline values of HED (through positive beliefs and wave 1 drinking frequency and quantity) and to the rate of growth in HED (through positive beliefs and wave 1 quantity). Adolescents with higher levels of depressive symptoms and positive alcohol beliefs drink more frequently and at greater quantities, which is associated with initiating HED at a higher level and escalating HED more rapidly than peers with similar depressive symptoms who lack those beliefs. This suggests that, to the extent that positive alcohol beliefs can be tempered through public health campaigns, education and/or counseling, HED among depressed adolescents might be reduced.

## 1. Introduction

Underage drinking, especially heavy episodic drinking (HED), is a worldwide public health concern. Some youth surveillance systems define HED as consumption of five or more standard drinks (i.e., containing 14 g of pure alcohol [1]) on a single occasion [2,3]. In the U.S. in 2019, 4% of 8th graders, 9% of 10th graders and 14% of 12th graders reported HED and 2–5% reported “extreme” HED (≥10 drinks) within the preceding two weeks [3]. Similarly, in Germany in 2016, 15% of 12–17-year-olds reported past-month HED, including 5% of 12–15-year-olds and 32% of 16–17-year-olds [4]. The 2015 prevalence of past-month HED among European 16-year-olds ranged from 8% in Iceland to 56% in Denmark [2]. In contrast to adults who typically drink more frequently but consume less, adolescents tend to consume more on fewer occasions, contributing to a variety of co-occurring problems with short- and long-term negative consequences [5]. Animal studies provide strong evidence that intermittent alcohol exposure in adolescence increases risks for adult psychopathology, including alcohol dependence, by altering neural circuits [6]. This is particularly worrisome given that adolescents who engage in HED are more likely than their peers to continue to do so at least through their early 40s [7].

Identified predictors of HED in adolescence include demographics (e.g., age, sex, socioeconomic status (SES)); social influences (e.g., peer and parental drinking, social norms); intrapersonal/psychological characteristics (e.g., sensation-seeking, impulsivity, depression, alcohol-related attitudes or expectancies); and environmental factors (e.g., exposure to alcohol marketing, availability of alcohol) [1,8]. Beliefs about alcohol’s effects develop in childhood, prior to experience with alcohol and based largely on social modeling, while experience with alcohol refines and hardens those beliefs [9]. Adoption of positive beliefs (i.e., that drinking would produce a positive outcome) increases between ages 8 and 14, slowing somewhat beyond age 12 [10]. Both age and experience with drinking are related to the strength alcohol beliefs. Stronger positive and weaker negative beliefs were held by 15-year-olds held than by 13-year-olds, and by 13-year-olds compared to 11-year-olds [11]. Moreover, regardless of age, heavy/binge drinkers held stronger positive beliefs than non-binge drinkers, whose positive beliefs were stronger than those of non-drinkers [11]. Finally, the belief that alcohol enhances relaxation and tension reduction grew rapidly among U.S. 5th and 6th graders (ages 10–11 and 11–12, respectively) followed over 18 months [12].

The relationship between children’s and adolescents’ positive alcohol beliefs and heavy drinking patterns is well-established. Smit et al. systematically reviewed 43 longitudinal studies examining the development, correlates, and predictive power of alcohol expectancies from age 4 to 18 years [9], of which 26 focused on beliefs as predictors of alcohol use. Positive beliefs were related to more drinking in all but two studies with younger samples [Smit]. More importantly, positive beliefs predicted onset and changes in heavier drinking in all five studies investigating that pattern [13,14,15,16,17]. For example, positive expectancies increased more rapidly among adolescents who began drinking early (i.e., ages 12–14) than among those who began at or after age 17, and adolescents with positive expectancies that alcohol could enhance their social relationships, moderate worries, make them feel good and forget problems reported having gotten drunk earlier and began binge drinking earlier than peers without such expectancies [17]. Similarly, compared to peers with fewer positive beliefs/expectancies about alcohol, adolescents who held more positive beliefs prior to age 16 drank more, increased their drinking more between ages 16 and 35, and were more likely to engage in alcohol misuse at age 35 [18]. Expecting positive experiences guides decisions about engaging in risky behavior [19] so, to the extent that positive expectancies can be reduced, risky drinking might also be reduced.

Young people also differ in their motives to drink alcohol [20]; most are motivated to enhance positive emotions and/or facilitate social connections, and some to cope with (i.e., change or reduce) negative emotional states. Two developmental pathways appear to explain the association between early risk factors and later disorders [21]. The “externalizing” pathway comprises behaviors and traits characterized by behavioral disinhibition, such as sensation-seeking and rebelliousness, antisocial behavior and attention deficit hyperactivity disorder, and the “internalizing” pathway reflects difficulties in dealing with negative affect and comprises behaviors and traits such as depression and anxiety. While the pathways link exposure to risk factors to development of HED, encountering protective factors may also prevent such development [21]. In line with the hypothesized internalizing pathway, Kuntsche et al. concluded that social motives are empirically related to moderate drinking, enhancement motives with heavy drinking, and coping motives with alcohol-related problems. In addition, the relationship between enhancement motives and heavy drinking was partially mediated by coping motives [20]. Hence, trying to change negative emotional states with alcohol seems to be the most problematic reason to drink, and adolescents with depressive symptoms might be especially vulnerable to this motive.

To our knowledge, the relationships among alcohol-related motives and beliefs, depressive symptoms and HED have not yet been investigated among youth. Our objective in this study was to examine associations between positive alcohol beliefs, depressive symptoms and HED in a population-based sample of adolescents in Germany. We hypothesized that depressive symptoms, drinking to relieve negative feelings, and positive beliefs about alcohol’s effects would be positively correlated with HED; further, that both drinking for relief and positive beliefs would mediate the relationship between depressive symptoms and HED.

## 2. Materials and Methods

### 2.1. Study Design and Participants

The study sample was recruited in three German states (Brandenburg, Hamburg and Schleswig-Holstein). From 744 schools on state listings, 120 schools were randomly drawn and invited for participation in May 2008. The randomization was stratified by school type and study region to ensure adequate representation of grade levels and student academic abilities and interests. After a 4-week recruitment interval, 29 schools agreed to participate and self-selected 176 6th–8th grade classes, with a total of 4195 students. School type composition of participating schools did not differ systematically from that of non-participating schools or from the official school statistics (χ^2^(3) = 0.74; n.s.). At baseline we surveyed 3414 students (81.4% of those sampled) from 174 classes; 134 students in 2 classes were absent and 646 were lacking parental consent. Follow-up included 3027 (88.6%) at nine months (wave 2), 1606 (47%) at 20 months (wave 3), and 1318 (38.6%) at 30 months (wave 4). Attrition was related to study region (fewer students from West Germany continued (*p* = 0.006)), and to lower SES, school performance and frequency of parental alcohol use (all *p*-values < 0.001). Attrition was also positively related to age, sensation-seeking/rebelliousness, past-month drinking and HED, favorable attitudes toward alcohol, friends’ drinking, and TV screen time (all *p*-values < 0.001).

### 2.2. Survey Administration and Data Collection

Parental consent forms were distributed by teachers three weeks prior to baseline data collection. Anonymous questionnaires were administered by trained personnel and self-completed during one school period; all students were free to refuse participation, though none did. Linking between baseline and follow-up questionnaires was facilitated by a student-generated anonymous 7-digit code [22].

### 2.3. Measures

Depressive symptoms (exposure variable) were assessed at baseline with the Depressive Symptoms Scale [23], which, when used with adolescents, is unidimensional in confirmatory factor analysis, correlates strongly with the depression scale of the Symptom Checklist 90 (SCL-90) and correlates more strongly than the SCL-90 with a diagnosis of major depressive disorder [23]. The scale is comprised of six items: “During the past three months, how often have you…felt too tired to do things?…had trouble going to sleep or staying asleep?…felt unhappy, sad, or depressed?…felt hopeless about the future?…felt nervous or tense?…worried too much about things?” Response options, never, rarely, sometimes, and often were scored 0 to 3, summed and averaged, yielding a scale ranging from 0 to 3 (α = 0.76). In the original version of the DSS, Kandel and Davies used three response categories (not at all, somewhat, much), scored 1–3, then averaged over the six items and multiplied by 10 to yield a scale with a range from 10–30 [23].

At each wave, students who answered “yes” to “Have you ever drunk alcohol?” were asked three questions: Frequency was assessed with “How often do you currently drink alcohol?” (0 = Never, 1 = Less than once a month, 2 = At least once a month, but not every week, 3 = At least once a week, but not daily, 4 = Daily) [24]. Quantity was assessed separately for beer, wine and spirits with “On the last day that you drank, how much did you drink?” In waves 1 and 2 (0 = 0 drinks, 1 = Less than 1 drink, 2 = 1–2 drinks, 3 = 3–4 drinks, 4 = 5 or more drinks); in waves 3 & 4 (4 = 5–6 drinks; 5 = >6 drinks). Quantities were then combined across alcohol types [2]. HED (the outcome measure) was assessed with “How often have you had 5 or more drinks on one occasion?” (0 = Never, 1 = Once, 2 = 2 to 5 times, 3 = More than 5 times) [3,25].

Drinking to relieve negative feelings was assessed with “When you feel stressed, do you want to drink alcohol?” (no/yes). We assessed the validity of this item by comparing the proportion of endorsers and non-endorsers who reported engaging in HED more than once in waves 2–4. Those who endorsed the drinking for relief item at wave 1 were more likely than non-endorsers to have engaged in HED more than once at each subsequent wave—42% vs. 11% at wave 2; 63% vs. 26% at wave 3; and 76% vs. 39% at wave 4 (all *p* < 0.001). Alcohol beliefs were assessed by rating four statements that alcohol (a) is relaxing, (b) makes you more outgoing, (c) brings a good mood, and (d) is something positive, on a scale from 0 = Not true at all to 3 = Totally true [26]. Responses were summed and averaged (α = 0.80).

Covariates included age, sex, SES [27], state of residence, and school type (4 types, including primary, trade, vocational/general education and university-preparation); sensation-seeking/rebelliousness [28]; and environmental influences: frequency of each parent’s drinking (0 = Never, 1 = Rarely, 2 = Often, but not daily, 3 = Daily) and number of drinking friends (0 = None, 1 = Some, 2 = Most, 3 = All). For analysis we divided parental drinking into never/rarely vs. often/daily and drinking friends into none vs. any. Sensation-seeking/rebelliousness was assessed with four items [29]: “I get in trouble in school”; “I do things my parents wouldn’t want me to do”; “I like scary things”; and “I like to do dangerous things”. Response options were the same as those for alcohol beliefs, scored 0–3, summed and averaged (α = 0.73).

### 2.4. Data Analyses

To ensure that we were examining the relationship between depressive symptoms and the progression of HED, we limited our analyses to adolescents who had engaged in HED once or never at wave 1. Preliminary analyses were conducted with Stata versions 14.2 (Revised 2018) and 15.1 (Revised 2018; Stata Corp., College Station, TX, USA), and latent growth modeling was performed in Mplus (version 7; Muthén & Muthén, 1998–2012).

To test our focal hypotheses, we used latent growth modeling with robust full information maximum likelihood estimation (MLR), an approach that both provides robust estimates when data are missing at random and allows for non-normality in the data [30,31]. In step 1, an unconditional model was fitted for HED to permit a test of change trajectories (i.e., linear versus quadratic) and to identify between-person differences in initial levels of HED (i.e., intercepts) and the rates of change over time (i.e., slopes). The model specified a random latent intercept (λ = 1) and random linear slope (λ = 0, 1, 2, 3). After inspecting model fit indices and the mean and variance of the intercepts and linear slopes, we then added a random latent quadratic slope (λ = 0, 1, 4, 9) to determine whether the rate of growth in HED changed over time. The best fitting model (i.e., linear vs. quadratic) was retained. A non-significant MLRχ^2^ was interpreted as good data-model fit. However, given the sensitivity of MLRχ^2^ to sample size, we also interpreted other fit indices [32]. In step 1 model fit was deemed acceptable if comparative fit index (CFI) ≥ 0.90 and root mean square error of approximation (RMSEA) ≤ 0.08 [33]. Where appropriate, acceptable models were compared using the Bayesian Information Criterion (BIC) with a smaller BIC indicating a better fitting model [34].

In step 2, a conditional parallel process latent growth model was tested [35]. Depression, alcohol beliefs, drinking for relief, and frequency and quantity of alcohol consumption were included as direct and indirect predictors of the intercept and slope of HED. We controlled for baseline age, sex, SES, state, school type, sensation-seeking/rebelliousness, frequency of father’s drinking, frequency of mother’s drinking, and friends who drink. In step 2, model fit was judged good if RMSEA and the upper bound of its 90% confidence interval (CI) were <0.05, CFI and TLI were ≥0.95, and SRMR was <0.05 [32]. In the final model, bootstrapped confidence intervals were calculated with 2000 replications [36].

## 3. Results

### 3.1. Participants Retained vs. Not Retained for Analyses

Table 1 presents baseline characteristics of the 3021 participants retained for analyses and the 393 who were not retained because they had either engaged in HED more than once at wave 1 or were missing data on HED in wave 1. Retained participants were younger, more likely to be female, reside in Schleswig-Holstein, attend university-preparation schools, have higher SES, and report doing well in school than those not retained. They reported less intense depressive symptoms and were less likely to smoke, to have parents who drank frequently or to have friends who drank, to hold positive beliefs about alcohol or to report drinking to relieve stress. They scored lower in sensation-seeking/rebelliousness, characteristics that are consistent with a lower probability of engaging in HED.

### 3.2. Missing Data

At wave 1, 90% of participants had complete data on all variables, 4.5% were missing only alcohol beliefs, which had the most missing values (135), and 2% were missing only father’s drinking. Participants missing data on alcohol beliefs were more likely to be male and less likely to drink frequently or in large quantities. In waves 2–4 missing values on the three outcome variables ranged from <0.1% to 2.8%. The 1102 participants (36.5%) who left the study after completing fewer than three waves were more likely to be younger and male, to attend trade/vocational or general education schools, to be higher in sensation-seeking/rebelliousness, and to drink less frequently than their peers who remained in the cohort.

### 3.3. Associations among Variables and Trajectories of Drinking

Zero-order correlations among alcohol beliefs, depressive symptoms, frequency, quantity and HED at wave 1 were all positive, moderate and statistically significant (Table 2), confirming expected relationships among the variables of interest. Frequency and quantity of drinking and number of HED occasions all increased over time (Table 3). At wave 1, 0.9% of participants drank at least weekly, increasing to 13.8% at wave 4. Similarly, 2.7% of participants reported consuming ≥3 drinks at wave 1, compared to 37.3% at wave 4. Finally, at wave 1 no participants had engaged in HED more than once; by wave 4, 40.3% reported having done so, with 16.4% having done so at least five times.

### 3.4. Unconditional Latent Growth Models

Table 4 presents fit indices for all models. Except for a significant MLRχ^2^, the unconditional latent growth model for HED demonstrated acceptable fit. The significant mean of the intercept indicated that the average baseline value of HED was different from 0 (*M*_i_ = 0.13 (SE = 0.006), *p* < 0.001), and the significant intercept variance (*D*_i_ = 0.11 (SE = 0.008), *p* < 0.001) indicated that participants differed with respect to these baseline values. The mean of the linear slopes indicated significant average growth in HED over time (*M*_s_ = 0.32 (SE = 0.008), *p* < 0.001), and the significant variance of the linear slopes (*D*_s_ = 0.12 (SE = 0.006), *p* < 0.001) indicated substantial between-person variability. Specifically, not all participants followed the same trajectory of HED over time.

A random latent quadratic slope was then added but did not improve the model. The mean quadratic slope (*M*_s_ = 0.01 (SE = 0.007), *p* = 0.15) was not significantly different from 0, despite a significant variance (*D*_s_ = 0.042 (SE = 0.006), *p* < 0.001). Thus, the unconditional model of HED containing a random intercept and random linear slope was preferred. Based on modification indices in the unconditional model, correlations among HED scores at wave 2 with both waves 3 and 4 were identified to improve model fit and were added to attenuate model misspecification. The MLRχ^2^ was reduced but remained significant. As other indices supported the fit of this model, it was retained and used to fit a conditional parallel process model.

### 3.5. Conditional Parallel Process Model

The conditional model included depressive symptoms, drinking for relief, alcohol beliefs, and frequency and quantity of drinking at wave 1, and the associations among them (see Figure 1), as well as covariates in each regression equation. The model estimated correlations between father’s and mother’s drinking, drinking for relief and alcohol beliefs, and quantity and frequency of drinking. Except for a significant MLRχ^2^ the model fit well and predicted 41.7% of the variance in the intercept and 32.8% of the variance in the slope of HED. The intercept and slope of HED were negatively correlated (r = −0.19, *p* = 0.02) indicating that participants who started higher in HED showed shallower growth trajectories in HED over time, likely reflecting a ceiling effect. Table 5 presents standardized regression coefficients and 95% CIs for the direct and indirect associations among the key variables.

#### 3.5.1. Depressive Symptoms and HED

Higher levels of depressive symptoms were indirectly associated with higher baseline values (intercept) of HED (*β* = 0.019, *p* = 0.001), with associations through alcohol beliefs and frequency of drinking (*p* = 0.002) and alcohol beliefs and quantity of drinking (*p* < 0.001). A direct relationship was not observed (*β* = 0.001). Similarly, higher levels of depressive symptoms were indirectly related to the rate of growth (slope) in HED (*β* = 0.009, *p* = 0.004), with associations through alcohol beliefs and quantity of drinking (*p* = 0.001). A negative direct relationship was observed (*β* = −0.032, *p* = 0.035), likely reflecting a ceiling effect, given that having more symptoms was related to starting HED at a higher level.

#### 3.5.2. Positive Beliefs, Drinking Motives and HED

Contrary to our expectations, neither alcohol beliefs nor drinking for relief was directly associated with the intercept or slope of HED; however, both were indirectly associated. Positive beliefs were related to the HED intercept through both frequency and quantity of drinking at wave 1, and to the HED slope through quantity of drinking at wave 1. Drinking for relief was related to both intercept and slope of HED only through quantity of drinking at wave 1 (Table 5).

#### 3.5.3. Positive Beliefs, Drinking Motives and Frequency and Quantity of Drinking at Wave 1

Positive beliefs and drinking for relief were robustly associated with both the frequency and quantity of drinking at wave 1 (Table 5).

#### 3.5.4. Other Predictors of HED

As expected, higher values for frequency and quantity of drinking at wave 1 were directly associated with higher baseline values for HED. Additionally, SES and sensation-seeking/rebelliousness were positively associated, and father’s drinking was negatively associated with the HED intercept. State of residence and school type were also related to baseline values for HED. Although drinking more per occasion (i.e., quantity) at baseline was positively associated with faster growth in HED, more frequent drinking was not. Covariates positively associated with the slope include sex, age, sensation-seeking/rebelliousness and having friends who drink, while SES was negatively associated with the HED slope. State of residence was also related to growth in HED.

## 4. Discussion

In this study we followed a large cohort of German adolescents over a period of two and one-half years to investigate progression of alcohol use and HED. At baseline, we found several known interrelations between participant characteristics and alcohol use (e.g., age, sensation seeking, friends’ drinking, and having favorable attitudes/beliefs towards alcohol). We also found less established relationships in this cohort, such as between alcohol use and depressive symptoms. Furthermore, we surprisingly found small but significant correlations between depressive symptoms and both positive attitudes/beliefs (e.g., that alcohol makes you feel better or makes you more outgoing) and drinking to relieve stress. To explore the interrelatedness of the study variables more deeply, and to shed further light on the role of depressive symptoms in the progression of alcohol use in youth, we built a conditional parallel process model with HED at each wave as the central outcome.

One novel finding from this study is that, after controlling for externalizing tendencies, adolescents with higher levels of depressive symptoms who also have more favorable beliefs about alcohol drink more frequently and at greater quantities, which in turn is associated with initiating HED at a higher level and escalating HED more rapidly than peers with similar depressive symptoms who do not hold those beliefs. Our central hypothesis was partially supported: positive alcohol beliefs, but not coping motives, mediated the relationship between depressive symptoms and the initial level of HED and partially mediated the relationship with the rate of change of HED.

Studies investigating the association between depressive symptoms or depressive disorder and alcohol use or alcohol use disorder in adolescents have so far yielded mixed results; 10/32 effects showed a positive association and 22 were null [37]. Our findings align particularly with those of Pesola et al., who found that, after controlling for externalizing behaviors, depressive symptoms at age 14 predicted “harmful drinking” (i.e., without parental permission, having whole drinks, number of whole drinks within 24 h, and having ever been drunk) at age 16 [38]. It is conceivable that one reason for null findings is failure to account for the influence of alcohol beliefs in depressed adolescents. Another factor which needs to be considered is the cultural context of the studies. While we assume a certain degree of universalism in the described relationships and mechanisms, prevalence of binge drinking in youth varies across countries, as do age of initiation, and norms and attitudes related to alcohol use. Favorable beliefs about alcohol are widely expressed in German society, which might encourage depressed youth to assume alcohol might be helpful to cope with the condition.

Limitations of this study include self-selection of participating schools and loss to follow-up of 36% of the overall sample, both of which might affect generalizability of our findings. The drop-out was mostly related to organizational reasons (e.g., school and class changes of students), however, the attrition analysis indicated that study drop-out was related to a number of individual risk markers, that is, SES, school performance, parental alcohol use, sensation-seeking/rebelliousness, past-month drinking and HED, favorable attitudes toward alcohol, friends’ drinking, and TV screen time. Hence, the results might be more indicative of students with a lower risk profile. Measures of drinking behavior were self-reported, although adolescents’ self-reports of drinking have been shown to reflect actual behavior [25]. Finally, although we accounted for a variety of factors related to drinking, other unmeasured variables might have influenced the associations we found. For example, there is some evidence that the relationship between alcohol expectancies and problematic drinking may be more nuanced in adolescents than in young adults. Social Learning Theory distinguishes between one’s beliefs about the likely outcome of a behavior (i.e., expectancies) and one’s confidence that one can control that behavior (i.e., self-efficacy). In one study of 14-year-olds, higher positive alcohol expectancies at time 1 predicted frequency, quantity and harmful drinking (measured by the Alcohol Use Disorders Identification Test) at time 2, but this association was fully mediated by higher drinking refusal self-efficacy [39].

## 5. Conclusions

From a clinical or school health perspective, the results suggest that, to the extent that positive alcohol beliefs can be tempered through public health campaigns, education and/or counseling, HED among depressed adolescents might be preventable with universal measures. Other studies reveal that positive expectancies increase more rapidly among adolescents who begin drinking early (i.e., ages 12–14) than among those who begin at or after age 17, and adolescents with positive expectancies that alcohol could enhance their social relationships, moderate worries, make them feel good and forget problems reported having gotten drunk earlier and began binge drinking earlier than peers without such expectancies [30]. This suggests that early intervention would likely pay the greatest dividends.

## Figures and Tables

**Figure 1 children-09-00103-f001:**
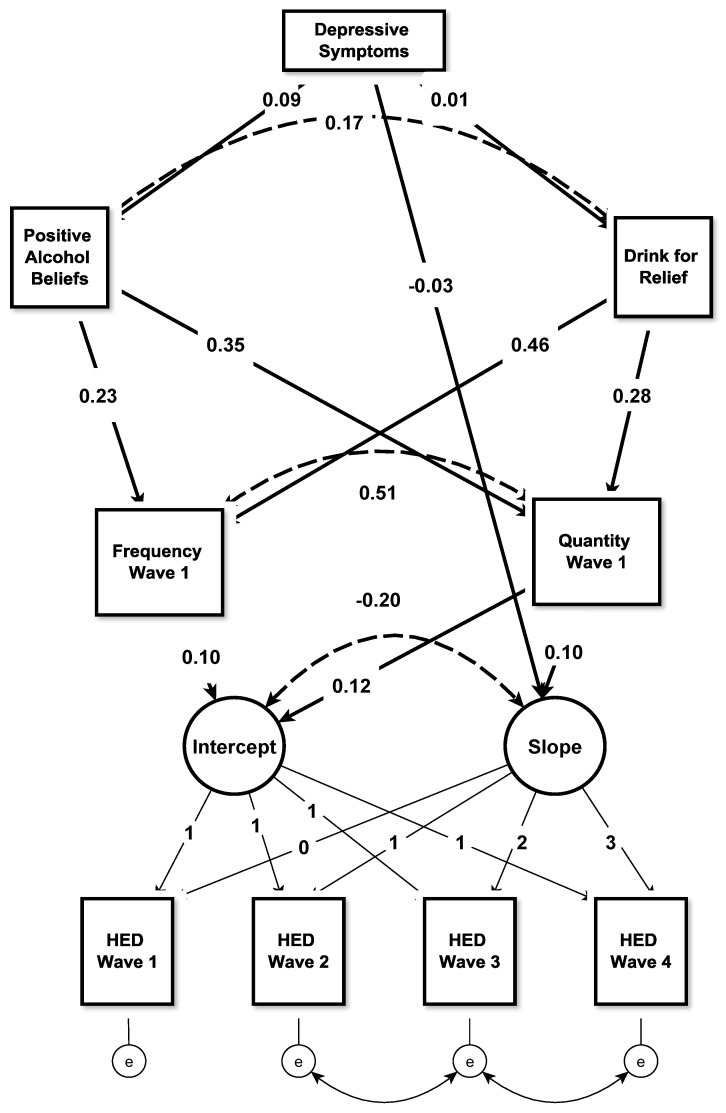
Parallel process linear growth model of the relationship between depressive symptoms and heavy episodic drinking (HED) over 2.5 years among adolescents who had engaged in HED never or only once at wave 1 (*n* = 3021). Covariates (age, sex, socioeconomic status (SES), state, school type, sensation-seeking/rebelliousness, parental and friends’ drinking) were included in all equations. Solid lines represent regressions; dotted lines represent correlations.

**Table 1 children-09-00103-t001:** Baseline characteristics of participants retained and not retained for analyses.

Study Variables	Retained (*n* = 3021)	Not Retained (*n* = 393)
Age, Mdn (IQR)	12 (12, 13)	13 (13, 14) ***
Sex, % female	52.7	44.1 ***
State, % Schleswig-Holstein	38.1	31.0 **
School type, % College preparatory	46.4	21.6 ***
Socioeconomic status, Mdn (IQR)	1.0 (0.7, 1.3)	0.7 (0.7, 1.0) ***
School grades, % good/very good	59.1	32.6 ***
Depressive symptoms, M (SD)	0.9 (0.5)	1.2 (0.6) ***
Positive beliefs about alcohol, Mdn (IQR)	0.0 (0.0, 0.8)	1.5 (0.8, 2.0) ***
Drink to relieve stress, % yes	2.3	25.1 ***
Father’s drinking, % more than seldom	30.1	38.3 ***
Mother’s drinking, % more than seldom	11.2	16.2 **
Any friends drink, %	42.0	89.1 ***
Sensation-seeking/Rebelliousness, Mdn (IQR)	0.5 (0.3, 1.0)	1.3 (0.8, 1.8) ***
Smoking frequency, % weekly/daily	30.2	69.8 ***

Mdn = median, M = mean, IQR = interquartile range, ** *p* < 0.01; *** *p* ≤ 0.001.

**Table 2 children-09-00103-t002:** Zero-order correlations ^a^ of key variables.

Variable by Number	1	2	3	4	5	6	7	8	9	10	11	12	13
1. Depressive symptoms	1.00												
2. Positive alcohol beliefs	0.24	1.00											
3. Drink to relieve stress	0.38	0.69	1.00										
4. Frequency wave 1	0.19	0.44	0.55	1.00									
5. Quantity wave 1	0.20	0.48	0.64	0.70	1.00								
6. HED wave 1	0.17	0.41	0.16	0.48	0.63	1.00							
7. Age	0.10	0.29	0.33	0.30	0.34	0.28	1.00						
8. Sex	−0.23	0.01	−0.02	0.02	0.06	0.04	0.05	1.00					
9. SES	0.02	−0.03	−0.29	−0.04	−0.06	−0.28	−0.14	−0.03	1.00				
10. Sensation-seeking/Rebelliousness	0.28	0.32	0.52	0.33	0.41	0.43	0.14	0.27	−0.09	1.00			
11. Father’s drinking	0.14	0.16	0.04	0.16	0.19	0.04	0.07	0.06	0.02	0.16	1.00		
12. Mother’s drinking	0.18	0.22	0.00	0.20	0.24	0.02	0.07	−0.01	0.19	0.17	0.36	1.00	
13. Friends’ drinking	0.24	0.43	0.14	0.40	0.54	0.27	0.48	0.03	−0.15	0.35	0.13	0.17	1.00

^a^ point-biserial coefficient Phi for dichotomous variables; Somers’ D for ordinal variables; Spearman’s Rho for ordinal variables.

**Table 3 children-09-00103-t003:** Frequency and quantity of drinking and heavy episodic drinking (HED) over data collection waves.

		Wave 1	Wave 2	Wave 3	Wave 4
Frequency of drinking, *n* (%)	Never	2322 (77.0)	1554 (59.2)	646 (40.6) ^a^	538 (31.9)
	<1/month	534 (17.7)	868 (33.1)	456 (28.7)	435 (25.8)
	<1/week	134 (4.4)	158 (6.0)	359 (22.6)	481 (28.5)
	>1/week & <daily	23 (0.8)	35 (1.3)	125 (7.9)	225 (13.4)
	Daily	2 (0.1)	11 (0.4)	5 (0.3)	7 (0.4)
Quantity of drinking (number of drinks) ^b^, *n* (%)	0	1764 (58.8)	1214 (45.4)	542 (34.2)	409 (24.2)
	<1	805 (26.9)	681 (25.4)	325 (20.5)	275 (16.3)
	1–2	350 (11.7)	457 (17.1)	341 (21.5)	373 (22.1)
	3–4	71 (2.4)	186 (7.0)	202 (12.7)	309 (18.3)
	≥5	8 (0.3)	139 (5.2)		
	5–6			99 (6.2)	174 (10.3)
	>6			77 (4.9)	147 (8.7)
HED frequency, *n* (%)	Never	2620 (86.7)	1996 (74.6)	907 (57.1)	773 (45.8)
	1 time	401 (13.3)	363 (13.6)	249 (15.7)	236 (14.0)
	2–5 times		217 (8.1)	267 (16.8)	403 (23.9)
	>5 times		101 (3.8)	166 (10.5)	277 (16.4)

^a^ Sum of proportions may differ from 100% because of rounding. ^b^ There were 5 categories for number of drinks at waves 1 and 2, and 6 categories at waves 3 and 4.

**Table 4 children-09-00103-t004:** Goodness-of-fit statistics for structural equation models of the relationship between depressive symptoms and HED.

Model	χ^2^ (df)	RMSEA (95% CI)	CFI	TLI	SRMR	BIC
1: ULG, linear slope	88.11 (5) ***	0.07 (0.06, 0.08)	0.93	0.92	0.021	16,586.36
2: ULG, linear & quadratic slopes	25.04 (1) ***	0.09 (0.06, 0.12)	0.99	0.88	0.023	16,642.59
3: Model 1, HED @ waves 2–4 correlated	26.97 (3) ***	0.05 (0.04, 0.07)	0.98	0.96	0.023	
4: CPP based on model 3	314.14 (47) ***	0.044 (0.039, 0.048)	0.96	0.91	0.037	

BIC = Baysean Information Criterion; CFI = comparative fit index; CPP = conditional parallel process model; 95% CI = 95% confidence interval; RMSEA = root mean square error of approximation; SRMR = standardized root mean square residual; TLI = Tucker-Lewis index; ULG = unconditional latent growth model; *** *p* < 0.001.

**Table 5 children-09-00103-t005:** Standardized regression coefficients (95% confidence intervals (CI) ^a^) from linear model of the association between depressive symptoms and heavy episodic drinking (HED) in adolescents who had engaged in HED ≤ 1 time at wave 1 (*n* = 3021).

Paths	Beta (95% CI) ^a^
Depressive symptoms -> HED Intercept	
Direct	0.003 (−0.045, 0.045)
Indirect	**0.019 (0.009, 0.032)** ^b^
Depressive symptoms -> Positive Alcohol Beliefs -> Intercept	0.003 (−0.001, 0.009)
Depressive symptoms -> Positive Alcohol Beliefs -> Frequency -> Intercept	**0.004 (0.002, 0.007)**
Depressive symptoms -> Positive Alcohol Beliefs -> Quantity -> Intercept	**0.008 (0.004, 0.013)**
Depressive symptoms -> Drink for Relief -> Intercept	0.002 (−0.001, 0.008)
Depressive symptoms -> Drink for Relief -> Frequency -> Intercept	**0.001 (0.000, 0.003)**
Depressive symptoms -> Drink for Relief -> Quantity -> Intercept	**0.001 (0.000, 0.002)**
Depressive symptoms -> HED Slope	
Direct	**−0.051 (−0.099, −0.002)**
Indirect	**0.009 (0.004, 0.017)**
Depressive symptoms -> Positive Alcohol Beliefs -> Slope	0.003 (−0.001, 0.010)
Depressive symptoms -> Positive Alcohol Beliefs -> Frequency -> Slope	0.000 (−0.002, 0.001)
Depressive symptoms -> Positive Alcohol Beliefs -> Quantity -> Slope	**0.005 (0.003, 0.009)**
Depressive symptoms -> Drink for Relief -> Slope	0.000 (−0.003, 0.003)
Depressive symptoms -> Drink for Relief -> Frequency -> Slope	0.000 (−0.001, 0.000)
Depressive symptoms -> Drink for Relief -> Quantity -> Slope	**0.001 (0.000, 0.002)**
Positive Alcohol Beliefs -> HED Intercept	
Direct	0.032 (−0.021, 0.067)
Indirect	**0.139 (0.113, 0.169)**
Positive Alcohol Beliefs -> Frequency -> Intercept	**0.047 (0.029, 0.067)**
Positive Alcohol Beliefs -> Quantity -> Intercept	**0.092 (0.067, 0.118)**
Positive Alcohol Beliefs -> HED Slope	
Direct	0.041 (−0.012, 0.100)
Indirect	**0.061 (0.044, 0.083)**
Positive Alcohol Beliefs -> Frequency -> Slope	−0.001 (−0.017, 0.015)
Positive Alcohol Beliefs -> Quantity -> Slope	**0.062 (0.048, 0.085)**
Drink for Relief -> HED Intercept	
Direct	0.014 (−0.012, 0.100)
Indirect	**0.061 (0.044, 0.083)**
Drink for Relief -> Frequency -> Intercept	−0.001 (−0.007, 0.015)
Drink for Relief -> Quantity -> Intercept	**0.062 (0.043, 0.085)**
Drink for Relief -> HED Slope	
Direct	0.000 (−0.049, 0.044)
Indirect	**0.012 (0.002, 0.023)**
Drink for Relief -> Frequency -> Slope	0.000 (−0.009, 0.007)
Drink for Relief -> Quantity -> Slope	**0.012 (0.005, 0.021)**

^a^ CIs are based on 2000 bootstrap replications. ^b^ Coefficients and CIs in **bold type** are statistically significant at *p* < 0.05. NOTE: Arrows (->) between constructs indicate paths in the model.

## Data Availability

The data presented in this study are available on request from the corresponding author. The data are not publicly available because parents did not consent to public use.

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
