# Peer review of "Depressive Symptoms, Alcohol Beliefs and Heavy Episodic Drinking in Adolescents"

_children, 2022, doi:10.3390/children9010103_

Round 1
Reviewer 1 Report
- In the research purpose section, author has stated that "To our knowledge, the relationships among alcohol-related motives and beliefs, depressive symptoms and HED have not yet been investigated among youth." I do not necessarily agree with this statement. It may be true that alcohol related beliefs, depression, and HED all together had not been examined in many studies, but there are number of research on alcohol related beliefs and heavy episodic drinking. I would recommend the researcher to conduct a more thorough literature review on their research topic.
- Please mention the specific sampling method in the materials and methods section
- What are the original sources of the measurements? Are the original measurement tools used in the survey or are they modified/revised for the current study? Provide more detailed descriptions of each measurements.
- Reorganize Table 5, it is hard to follow the paths at this point.
- Since the study participants are the German adolescents, more discussion and implication should be made in the context of German adolescents.
Reviewer 2 Report
The goal of this manuscript was to determine the associations between depressive symptoms, reasons for alcohol use (i.e. coping), anticipated effects of alcohol (i.e. positive effects), and heavy episodic drinking. The authors hypothesized these variables would be positively associated with heavy episodic drinking even after accounting for other factors (e.g. parental/peer influence, sensation seeking). Strengths of this study include the longitudinal design and the well-characterized sample.
Was information about other substance use available? Are these associations specific to alcohol or do they extend to other substances (i.e. cigarettes, marijuana). Are the current results significant after accounting for other drug use?
Table 2 reports correlations for HED wave 1; however, this is a binary variable based on inclusion exclusion criteria. Would another statistical test be more appropriate (same concern for any other binary variables in the table)? Also unclear what the indentation pattern in variable number/name means (column 1).
Table 1 should include depression symptoms. Furthermore, it would be useful to have some discussion about the mean and standard deviation in the sample. Are many participants likely to have clinical levels of depression or are most scores in the normative range?
Round 2
Reviewer 2 Report
The authors have addressed my comments